# Social Capital and HIV Testing Uptake among Indirect Female Sex Workers in Bali, Indonesia

**DOI:** 10.3390/tropicalmed5020073

**Published:** 2020-05-07

**Authors:** I Gusti Ngurah Edi Putra, Pande Putu Januraga

**Affiliations:** 1School of Health and Society, Faculty of Social Sciences, University of Wollongong, Wollongong, NSW 2522, Australia; ignep718@uowmail.edu.au; 2Center for Public Health Innovation, Faculty of Medicine, Udayana University, Denpasar, Bali 80361, Indonesia

**Keywords:** social capital, social cohesion, social participation, HIV testing, female sex workers, Indonesia

## Abstract

Indirect female sex workers (FSWs), a type of FSW working under the cover of entertainment enterprises (e.g., karaoke lounge, bar, etc.), remain as an important key population for HIV transmission, signaling the need of appropriate interventions targeting HIV-related behaviors. This study aimed to investigate the association between social capital and HIV testing uptake. A cross-sectional study was conducted among 200 indirect FSWs in Denpasar, Bali. The dependent variable was HIV testing uptake in the last six months preceding the survey. The main independent variables were social capital constructs: social cohesion (perceived peer support and trust) and social participation. Variables of socio-demographic characteristics were controlled in this study to adjust the influence of social capital. Binary logistic regression was performed. The prevalence of HIV testing in the last six months was 72.50%. The multivariate analysis showed that only peer support from the social capital constructs was associated with HIV testing uptake. Indirect FSWs who perceived a high level of support within FSWs networks were 2.98-times (95% CI = 1.43–6.24) more likely to report for HIV testing. Meanwhile, perceived trust and social participation did not show significant associations in relation to HIV testing uptake. As social cohesion (support) within FSWs’ relationships can play an important role in HIV testing uptake, existing HIV prevention programs should consider support enhancement to develop a sense of belonging and solidarity.

## 1. Introduction

The HIV epidemic in Indonesia remains concentrated among key populations such as female sex workers (FSWs), men who have sex with men (MSM), transgender people, people who inject drugs (PWID), and prisoners. Findings from the integrated biological and behavioral survey (IBBS) in 2015 showed that PWID and MSM were the most affected populations with HIV prevalence of 28.76% and 25.80%, respectively, followed by transgender people of 24.80% and FSWs of 5.30% [1,2]. Despite FSWs had the lowest HIV prevalence, they are thought to play important roles in HIV infection to the general population due to heterosexual intercourse is the main mode of HIV transmission in Indonesia [3]. The national estimation also suggested that FSWs and their clients contributed to 42–43% of new HIV infections during 2008–2014 [4], signaling FSWs are important key drivers of HIV transmission.

FSWs are commonly grouped as direct and indirect FSWs in Indonesia. Direct FSWs refers to those whose sex work is the main source of income and they provide sex services in red-light areas, commonly identified as brothel or street FSWs [5,6]. Meanwhile, indirect FSWs are those who sell sex services under the cover of recreational or entertainment enterprises that also indicates sex work is not their main source of income, such as FSWs in bars, karaoke, massage parlors, etc. [5,7]. The Integrated Biological and Behavioral Survey (IBBS) in 2015 confirmed that the proportion of indirect FSWs who have ever received HIV information and had comprehensive HIV knowledge was lower than their direct counterparts [1]. In addition, using the same dataset, we have reported only 50% of indirect FSWs have protected sex with clients in last month [8]. Lastly, indirect FSWs were 46% more likely to be infected with STIs [7] and only half of them (50.7%) were undergoing HIV testing, as reported by a study in Yogyakarta, Indonesia [9]. Obviously, low HIV testing rates among this key population also leads to low treatment uptake that decelerates the achievement of UNAIDS’ global 90-90-90 target by 2020 [10].

The concept of social capital is increasingly adopted in public health literature to explain social phenomenon underlying HIV-preventive behaviors among FSWs. A social capital theorist, Robert Putnam has conceptualized social capital as “the networks, norms, and social trust that facilitate co-operation for mutual benefit” [11]. His model allows social capital to be defined as two main types, namely bonding and bridging. Bonding refers to the relationship within-group or intra-group networks whereas bridging is defined as the interactions between groups of inter-group connection [11]. Those two main concepts are recently translated as social cohesion and social participation. Furthermore, Fonner et al. defined social cohesion as trust, mutual aids, and solidarity within FSWs community, whilst social participation refers to FSWs’ involvement in the wider community outside FSWs’ networks [12]. Social capital is an important component of community empowerment approaches for HIV prevention among FSWs [13,14] and has been identified to be associated with several HIV-related behaviors and outcomes among FSWs [12,13,15]. A study by Carrasco et al. found that a higher score of social cohesion was associated with consistent condom use with clients and can be a protective factor for STI infection among FSWs living with HIV in the Dominican Republic [13]. In addition, a study from Swaziland suggested that social cohesion was a predictor of consistent condom use and fewer reports of social discrimination, whilst social participation was a determinant for HIV testing and negatively associated with verbal and physical violence due to selling sex [12]. A previous quality study conducted in Bali, Indonesia also highlighted the importance of social capital between FSWs to build peer trust and solidarity that can lead to protected sex with clients [15]. Therefore, understanding the dynamics of social capital among FSWs potentially brings opportunities to set an appropriate intervention.

Denpasar, a capital city of Bali, is one of the Indonesian cities with a massive socio-economic improvement that attracts a lot of newcomers outside Bali to reside and look for sources of income. Therefore, the majority of direct and indirect FSWs in Denpasar are identified as non-Balinese [16]. A previous study confirmed the increase of HIV prevalence from 0% in 2000 to 7.2% in 2010 among indirect FSWs in Denpasar [5]. In addition, characteristics of indirect FSWs that have greater mobility and actively move from one to another venue might influence the social capital among them, in which, turns to influence their HIV-preventive behaviors, such as HIV testing uptake. Therefore, this study aimed to identify the association between social capital and HIV testing uptake among indirect FSWs in Denpasar, Bali.

## 2. Materials and Methods

### 2.1. Data and Samples

This was a cross-sectional study using the dataset from a project on “Social Capital Survey and Internet Utilization among Indirect Female Sex Workers”. Field data collection was carried out by outreach workers from the Kerti Praja Foundation (KPF), Bali from August to October 2017. The KPF is a well-known NGO in Bali with concerns on HIV and AIDS issues among key populations. Outreach workers from this NGO have long experiences in conducting outreach for FSWs. KPF also carries out six-monthly mapping to estimate the number of indirect FSWs working in several venues in Denpasar. Based on the last mapping prior to data collection of the project, there were 963 indirect FSWs estimated to be scattered in 102 venues. Six types of venue were identified that include bar, café, karaoke lounge, massage parlor, beauty salon, and spa. The primary data collection employed single-stage cluster random sampling by taking into account diverse number for each type of venue and distribution of indirect FSWs by venue. This aimed to ensure each category of working place was proportionally selected and well-represented. All indirect FSWs in selected venues were recruited as samples. The primary data collection successfully documented 200 indirect FSWs from 28 venues (1 bar, 15 cafés, 4 karaoke lounges, 6 massage parlors, 1 beauty salon, and 1 spa). The outreach workers used to conduct regular outreach activities in the same venues where data for the project were collected. This made them known by venue’s owners and FSWs and hence, most of the outreach workers did only one visit to the selected venues to interview indirect FSW. In case outreach workers could not meet indirect FSWs at the first visit due to some circumstances, more visits were conducted. Interviews using anonymous questionnaires were carried out in a private room available in venue settings. The recruitment of indirect FSWs was stopped when the desired sample size (200) was reached.

### 2.2. Variables

The dependent variable in this study was HIV testing uptake in last six months. Social capital as the main independent variable consisted of social cohesion and social participation. Following the definition of social cohesion provided by a previous study [12], social cohesion in this present study was measured as perceived trust and support independently. The questions for social capital constructs were adopted from some previous studies [12] and then translated into Indonesian language. The questionnaires have been tested among indirect FSWs in two venues prior to the main field data collection to ensure the questions were understandable and necessary revisions have been made. Indirect FSWs were asked to rate their agreement toward each statement of social cohesion and social participation. The exploratory factor analysis (EFA) was then used to identify the relationship among indicators and develop suitable indicators for two measurements of social cohesion (support and trust) and Cronbach’s alpha was applied to determine the reliability. One out of seven items of the perceived support was omitted due to low factor loading (below 0.3) and the remaining items showed acceptable internal consistency (α = 0.65). Meanwhile, all items of the perceived trust had medium-to-high factor loading and good reliability (α = 0.72). Furthermore, the score of all items was summed up and both perceived support and trust were classified as low (≤median score) and high (>median score).

Social participation was assessed using four items with “yes” and “no” responses on FSWs’ involvement in community groups outside of the FSW relationship. FSWs who participated in at least one out of four community groups were grouped as high social participation whereas low social participation was assigned to those who did not participate at all. The indicators used to measure social cohesion (support, trust) and social participation are listed in Table 1. In addition, socio-demographic characteristics of FSWs were also documented such as age, educational level, marital status, residential status, type of venues, and duration of sex work.

### 2.3. Data Analysis

Descriptive statistics was used to present the prevalence of HIV testing uptake, data distribution of socio-demographic characteristics, and social capital among indirect FSWs. In this study, the variables of social capital were treated as a binary variable to compare the odds of HIV testing between FSWs who were in a group of high perceived support, trust, or with social participation and their counterparts who were grouped with low support, trust, or no social participation, respectively. Binary logistic regression was applied for bivariate and multivariate analysis. Bivariate analysis was used to assess the effect of independent variables (e.g., social capital constructs, socio-demographic characteristics) individually in relation to HIV testing uptake. Meanwhile, multivariate analysis used the enter method to identify the adjusted association between social capital and HIV testing uptake after controlling all socio-demographic characteristics. The results were reported as odds ratio (OR), 95% confidence interval (CI) OR, and *p*-value.

### 2.4. Ethical Consideration

The protocol of primary data collection has obtained ethical approval from the Ethics Committee of Faculty of Medicine, Udayana University, Bali, Indonesia with No. 1094/UN.14.2/KEP/2017. Interviews were performed by professional outreach workers from the KPF who have been known by indirect FSWs. Data were collected anonymously to ensure the confidentiality and verbal informed consent was given by FSWs.

## 3. Results

In Table 2, we present the socio-demographic characteristics, social capital, and HIV testing uptake among indirect female sex workers (FSWs). Out of 200 indirect FSWs in this study, the majority aged 25 years and above (132; 66.00%) and have ever got married but not currently in the marital relationship (119; 59.50%). About half of the indirect FSWs completed senior high school (103; 51.50%) and only fewer FSWs (5; 2.50%) attended higher education. Based on residential status, almost all indirect FSWs were from outside Bali (190; 95.00%). There were some popular venues where FSWs worked at the time of the survey, such as karaoke and cafe (65; 32.50% for each). Meanwhile, less than 5% of them worked in the salon and bar, respectively. Almost three out of four indirect FSWs in Denpasar reported having HIV testing in the last six months (145; 72.50%). Based on social capital constructs, less than half of the indirect FSWs perceived a high-level of support from their internal networks (97; 48.50%) and highly trusted their peers (93; 46.50%). However, only 28.00% of them participated in groups, clubs, or organizations outside the FSWs relationship.

In bivariate and multivariate analysis, some categories of variables were collapsed together due to low percentages in some particular categories. For the educational level, the categories of senior high school and university were merged as senior high school and higher. Moreover, categories of the work venue were re-grouped into two (massage parlor, spa vs. salon karaoke, bar, café). Findings from bivariate analysis (Table 3) showed that indirect FSWs who were currently or ever married were less likely to report for having HIV testing in last six months compared to single ones, but the association was not statistically significant. In addition, indirect FSWs who completed junior high school or higher increased the likelihood of HIV testing compared to those who only completed primary school. Moreover, those who have engaged in sex work for more than three years were 2.10 times (OR = 2.10; 95% CI = 1.01–4.40) more likely to report for HIV testing compared to those working for one year or less. The odds of HIV testing were not different between those working in two main groups of venues (massage parlor, spa, and salon vs. karaoke, bar, and cafe). For social capital constructs, only peer support was associated with HIV testing. Those who perceived a high level of support were 2.75-times (OR = 2.75; 95% CI = 1.42–5.32) more likely to report for HIV testing uptake in last six months.

Multivariate analysis (Table 3) examined the adjusted effect of social capital after controlling all socio-demographic characteristics. Similar to the result of bivariate analysis, perceived support increased the likelihood of HIV testing by almost three times (OR = 2.98; 95% CI = 1.43–6.24). In addition, some socio-demographic characteristics turned to be significant predictors. Indirect FSWs who ever married were less likely to report HIV testing in last six months (OR = 0.37; 95% CI = 0.14–0.98). In addition, those who completed junior high schools were nearly four times (OR = 3.80; 95% CI = 1.17–12.35) more likely for undergoing HIV testing compared to those who completed primary school. Findings from multicollinearity testing suggest that none of the independent variables was found to be highly correlated (r < 0.5).

## 4. Discussion

This study found that social cohesion, particularly perceived high-level of support increased the HIV testing uptake by almost three-fold among indirect FSWs even after controlling all socio-demographic characteristics. A previous qualitative study suggested that social support within the FSWs network plays important roles as facilitators to testing [17]. Social support and existing networks among FSWs could also increase the acceptability of HIV testing [18]. Some previous studies suggested that motivation and encouragement from peers in FSWs relationships helped develop awareness and create demand for HIV services [19,20,21,22]. In the context of HIV prevention program in Denpasar, Bali, indirect FSWs from some particular venues were recruited as peer educators and trained by a local NGO in order to help their community to receive necessary HIV information and services. Several activities performed by trained peers in FSWs’ community potentially led to intense communication and interaction, and also developed social support that could help FSWs to not only deal with their daily life matters but also assist them to seek HIV services. Meanwhile, trust was not associated with undergoing HIV testing in this study. It might be due to some indicators for measuring the perceived trust in this study that were only related to general trust, not specifically assessing whether they can count on sex worker colleagues to access services and keep the unexpected result from HIV testing.

The low level of social participation among FSWs found in this study may be due to the socio-environmental situation of Bali which is different from the FSWs’ origin culture and values as the majority of them are not Balinese [15]. As internal migrants who have different cultural attributes from the receiving community, FSWs might tend to isolate themselves in FSWs’ community only and be reluctant to participate in large society outside of FSWs’ networks. In addition, FSWs are a marginalized population, as sex work remains illegal in Indonesia. As a consequence, they might face stigmatization, discrimination, and rejection, as well as the fear of being arrested by police if their status is disclosed [23,24], all of which contribute in preventing them from participating in social activities in the host area. Those situations also help explain why the finding of this study is inconsistent with a previous study that social participation was associated with HIV testing [12].

Other important findings from this study are that socio-demographic characteristics had apparent effects on HIV testing uptake among indirect FSWs. FSWs who completed junior high school were more likely to report for HIV testing in last six months preceding to survey compared to those who completed primary school only. A high level of education can contribute to sufficient knowledge on HIV [25], access to services such as knowing a place for HIV testing [26], and perceived of HIV risk [27] that increase the likelihood of undergoing HIV testing [28,29,30]. In addition, those who were divorced or widowed were less likely for being tested for HIV in last six months. It may be due to single FSWs having more concerns with their health compared to ever married FSWs, and hence, they tend to have higher uptake of HIV testing as one of the HIV-preventive behaviors. The findings from previous studies also confirmed that single FSWs were reported with higher rates of consistent condom use and also HIV-negative [31,32], indicating unmarried FSWs might have more concern in maintaining preventive behaviors. In some situations, having more concerns with unwanted pregnancy might partly explain why single FSWs are more likely to do HIV preventive behavior (e.g., protected sex with clients for contraceptive purpose) since premarital pregnancy is not accepted in Asian cultures [33,34]. Further, a case study carried out in Kolkata, India, showed that divorced women with the responsibility of raising children pushed them into sex work due to economic necessity and this pressing situation might complicate them from avoiding health risk behaviors as FSWs [35]. However, further investigation is needed to investigate the association between marital status and HIV preventive behavior. Moreover, despite that the variable of work duration for sex work was not statistically significant when associated with HIV testing, this study found that higher odds of HIV testing in accordance with a longer period engaged in sex work. Similar to previous studies, higher uptake of HIV testing was observed among those who have been in sex work for a longer period of time, which might be due to more exposure of information on HIV and accessible HIV services [27,36,37].

Findings from this study suggest that including the social cohesion (support) escalation in HIV prevention program among indirect FSWs in Denpasar could be beneficial. Peer support within FSWs networks, should be consistently improved and maintained through outreach activities performed by outreach workers from a local NGO. To date, the HIV prevention program in Denpasar has targeted the enabling environment of FSWs by training some indirect FSWs as peer educators. It aims to make them capable of providing necessary support for other FSWs (e.g., providing HIV information, bridging to HIV services). The presence of peer educators potentially fosters social support within their intra-group networks. Moreover, the intervention delivered by outreach workers should cover some activities aiming to increase the bonds and interaction among FSWs. For instance, providing space and time for gathering, sharing problems, and developing collective actions is substantial to increase the sense of community, ownership, and solidarity [38,39]. Furthermore, the Sonagachi model in India can be adopted to enhance social support among FSWs [40]. This is a community-led approach that involved multilevel interventions to reduce environmental barriers and create conducive atmospheres that support for HIV preventive behaviors. Within the supportive environment, activities for the enhancement of intra-group networks and improvement for individual skills and knowledge for HIV prevention are doable.

This study has some limitations. Firstly, the cross-sectional study design precludes drawing a conclusion on causality between social capital and HIV testing. Secondly, all measurements of variables in this study relied on self-reported responses that are potentially influenced by social desirability. In addition, the complex constructs of social capital were only measured as social cohesion (support and trust) and social participation in this study, and were not extensively tested for validity and reliability, and hence, the indicators might be not applicable in other settings.

## 5. Conclusions

HIV testing uptake among indirect FSWs in the last six months prior to the survey was at the moderate-to-high level (72.50%), but a noticeable proportion of them did not undergo HIV testing (27.50%). Social cohesion, particularly perceived peer support, appeared as a significant predictor of HIV testing by increasing the likelihood of HIV testing by almost three times. Therefore, enhancement of social support should be an integral part of the HIV prevention program by allowing FSWs to have space and time for developing a sense of belonging and solidarity within their community.

## Figures and Tables

**Table 1 tropicalmed-05-00073-t001:** The measurement of social capital constructs.

Variables	Items	Properties	Alpha
Social cohesion (Support)	Communication and interaction with other FSWs for these following activities: Giving a gift or exchange gifts Working or doing something together Looking for entertainment, playing, or traveling together Making a call via telephone or internet Aiding others Participating in FSWs’ groups or organizations *Response choices: never (0), seldom (1), almost every day (2), every day (3)*	Mean = 5.36 SD = 2.67 Median = 5	0.65
Social cohesion (Trust)	Agreement with these following statements: You believe that other FSWs support each other to use condoms FSWs’ group or network where you work is compact General speaking, you can believe in other FSWs General speaking, FSWs who are in the same area as you only care to themselves General speaking, FSWs who get along with you, always fight with each other *Response choices: strongly disagree (1), disagree (2), neutral (3), agree (4), strongly agree (5)*	Mean = 16.9 SD = 3.12 Median = 17	0.72
Social participation	Participation in these following groups: Affiliations with religious groups Affiliations with study or exercise clubs Cultural groups, such as dancing or music, etc. Community activities, such as PKK (family welfare), women’s empowerment groups, etc. *Response choices: yes (1) no (0)*	Mean = 0.41 SD = 0.76 Median = 0	-

**Table 2 tropicalmed-05-00073-t002:** Socio-demographic characteristics, social capital, and HIV testing uptake among indirect female sex workers (FSWs).

Variables		n = 200	%
**Socio-Demographic Characteristics**			
Age	min, max	18, 50	-
	<25 years old	68	34.00
	≥25 years old	132	66.00
Marital status	single	51	25.50
	currently married	30	15.00
	ever married (divorced/widowed)	119	59.50
Educational level	primary school	24	12.00
	junior high school	68	34.00
	senior high school	103	51.50
	university	5	2.50
Residential status	outside Bali	190	95.00
	Bali	10	5.00
Duration of sex work	min, max	<1, 16	-
	≤ 1 year	85	42.50
	>1–3 years	47	23.50
	> 3 years	68	34.00
Work venue	massage parlor	38	19.00
	spa	20	10.00
	salon	4	2.00
	karaoke	65	32.50
	bar	8	4.00
	cafe	65	32.50
**Social Capital**			
Peer support	low	103	51.50
	high	97	48.50
Trust	low	107	53.50
	high	93	46.50
Social participation	low	144	72.00
	high	56	28.00
HIV testing uptake	no	55	27.50
	yes	145	72.50

**Table 3 tropicalmed-05-00073-t003:** Bivariate and multivariate analysis of social capital and HIV testing uptake.

Variables		Bivariate Analysis		Multivariate Analysis	
		OR (95% CI)	*p*-Value	aOR (95% CI)	*p*-Value
**Socio-Demographic Characteristics**					
Age	< 25 years old	ref		ref	
	≥ 25 years old	1.15 (0.60–2.21)	0.664	1.80 (0.77–4.18)	0.172
Marital status	single	ref		ref	
	currently married	0.90 (0.31–2.65)	0.854	0.77 (0.21–2.84)	0.696
	ever married	0.61 (0.28–1.32)	0.209	0.37 (0.14–0.98)	0.045
Educational level	primary school	ref		ref	
	junior high school	2.80 (0.99–7.88)	0.051	3.80 (1.17–12.35)	0.027
	senior high school and higher	1.31 (0.52–3.28)	0.570	1.59 (0.54–4.73)	0.402
Residential status	outside Bali	ref		ref	
	Bali	0.36 (0.10–1.29)	0.115	0.28 (0.07–1.22)	0.090
Duration of sex work	≤1 year	ref		ref	
	>1–3 years	1.79 (0.80–4.00)	0.160	1.87 (0.76–4.58)	0.172
	>3 years	2.10 (1.01–4.40)	0.048	2.17 (0.93–5.04)	0.072
Work venue	massage parlor, spa, salon	ref		ref	
	karaoke, bar, cafe	0.88 (0.45–1.74)	0.719	0.81 (0.37–1.80)	0.604
**Social Capital**					
Peer support	low	ref		ref	
	high	2.75 (1.42–5.32)	0.003	2.98 (1.43–6.24)	0.004
Trust	low	ref		ref	
	high	0.87 (0.47–1.61)	0.651	1.00 (0.49–2.03)	0.996
Social participation	low	ref		ref	
	high	1.36 (0.67–2.79)	0.398	1.06 (0.47–2.40)	0.884

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
