# Peer review of "Social Capital and HIV Testing Uptake among Indirect Female Sex Workers in Bali, Indonesia"

_tropicalmed, 2020, doi:10.3390/tropicalmed5020073_

Round 1

Reviewer 1 Report

The authors present a generally clear and well-written manuscript that describes the association between social capital and HIV testing uptake. My main criticism is that way more details on data collection and statistical analysis need to be given and issues of multicollinearity addressed.

Major comments:

More details on the recruitment procedure need to be provided. How many visits to each venue? How many respondents from each venue? How many venues of the same type? How was the randomisation of vanues done (i.e. what constituted the full list of venues). Was the recruitment stopped after 200 people were reached?

Furthermore, where were the interviews conducted, how was privacy ensured? The latter significantly affects participant’s bias in the matters of social support.

Have you tested for multicollinearity in your analysis? Age and duration of sex work? Or age and being ever married?

Multiple comparisons adjustments need to be done in bivariate analysis.

Please explain the enter method used to add variables in the multivariate analysis. Right now it seems looks like all variables were added at once.

Minor comments:

Please, make sure you spell out abbreviations.

Please provide range for years and other continuous variables.

Explain which categories were further merged (line 127).

Please provide the OR and 95% CIs in the text (Results section)

Table 2 Marital status categories are hard to read as lines don’t match.

Reviewer 2 Report

This is an interesting article exploring potential predictors of HIV testing among female sex workers in Bali.  I have the following thoughts/comments.

Abstract:

  • The term “indirect FSW” needs to be clarified for those readers who are not as familiar with this field.
  • The final sentence should be modified to indicate the need for enhancement of social cohesion/support, not necessarily social capital in general.

Introduction:

  • I recommend a little bit more development of the discussion about why social capital may be particularly important among the FSW population. That is, why do we think this is particularly important to explore within this community?  Could this be a unique opportunity of where to intervene in an otherwise challenging group to reach?

Methods:

  • The first paragraph indicates that “all indirect FSWs” in selected venues were recruited. Given the challenge of reaching this group, it would be helpful to give some more details on how the group knew who “all FSWs” were in each venue (i.e., how was the sampling frame created) and then how did they go about reaching them.
  • Where were the social capital questions drawn from, and how were they validated within the Balinese population?
  • Given that the social capital variables were combined in different ways “i.e., low vs. high” the data analysis paragraph should clarify how the variable was used in the models.

Results:

  • I recommend giving the numbers as well as the percentages in the text.
  • In the 2nd paragraph, I would insert some specific odds ratios to highlight what is being brought out in the text.

Discussion:

  • In the first sentence, I would be more specific about social cohesion being the prominent factor, not all of the social capital construct.
  • There should be more clarity around the terms “social support” and “social cohesion” and “trust”, and how they are being used differently in this paper.
  • In the 3rd paragraph, it is unclear why the authors think that single FSWs care about their health more than those who had ever been married. I recommend being careful with such a broad-reaching conclusion. 
  • Given the challenge and stigma associated with being an FSW, how do the authors propose to encourage activities between them? Are there examples of such programs in the literature that they could draw from?
